# Genome-Wide Identification and Expression Analysis of the NLP Family in Sweet Potato and Its Two Diploid Relatives

**DOI:** 10.3390/ijms26178435

**Published:** 2025-08-29

**Authors:** Kui Peng, Wenbin Wang, Zhuoru Dai, Meiqi Shang, Hong Zhai, Shaopei Gao, Ning Zhao, Qingchang Liu, Shaozhen He, Huan Zhang

**Affiliations:** 1Frontiers Science Center for Molecular Design Breeding, Key Laboratory of Crop Heterosis and Utilization (MOE)/Key Laboratory of Sweet Potato Biology and Biotechnology, Ministry of Agriculture and Rural Affairs/Beijing Key Laboratory of Crop Genetic Improvement/Laboratory of Crop Heterosis & Utilization and Joint Laboratory for International Cooperation in Crop Molecular Breeding, Ministry of Education, College of Agronomy & Biotechnology, China Agricultural University, Beijing 100193, China; kuipeng2021@163.com (K.P.); 15703549362@163.com (W.W.); daizhuoru@cau.edu.cn (Z.D.); 15545251361@163.com (M.S.); zhaihong@cau.edu.cn (H.Z.); spgao@cau.edu.cn (S.G.); zhaoning2012@cau.edu.cn (N.Z.); liuqc@cau.edu.cn (Q.L.); 2Sanya Institute of China Agricultural University, Sanya 572025, China

**Keywords:** sweet potato, NLPs, expression analysis, nitrate signaling response

## Abstract

NIN-like proteins (NLPs) are conserved, plant-specific transcription factors that play crucial roles in the nitrate signaling response, plant growth and development, and abiotic stress responses. However, their functions have not been explored in sweet potato. In this study, we identified 7 *NLPs* in cultivated hexaploid sweet potato (*Ipomoea batatas*, 2n = 6x = 90), 9 *NLPs* in the diploid relative *Ipomoea trifida* (2n = 2x = 30), and 12 *NLPs* in *Ipomoea triloba* (2n = 2x = 30) via genome structure analysis and phylogenetic characterization, respectively. The protein physiological properties, chromosome localization, phylogenetic relationships, syntenic analysis maps, gene structure, promoter *cis*-acting regulatory elements, and protein interaction networks were systematically investigated to explore the possible roles of homologous *NLPs* in the nitrate signaling response, growth and development, and abiotic stress responses in sweet potato. The expression profiles of the identified *NLPs* in different tissues and treatments revealed tissue specificity and various expression patterns in sweet potato and its two diploid relatives, supporting differences in the evolutionary trajectories of the hexaploid sweet potato. These results are a critical first step in understanding the functions of sweet potato *NLPs* and offer more candidate genes for improving nitrogen use efficiency and increasing yield in cultivated sweet potato.

## 1. Introduction

Nitrogen (N), the main limiting factor for plant growth, is fundamental to agricultural productivity, animal and human nutrition, and sustainable ecosystems [1,2,3,4]. N is also an important component of biological macromolecules such as proteins, nucleic acids, coenzymes, and chlorophyll in plants [5]. The availability of N in agricultural fields significantly affects crop yields [6]. In plants, nitrogen uptake and utilization from the soil is mainly through two forms—nitrate (NO_3_^−^) and ammonia (NH_4_^+^). In terrestrial plants, nitrate is the main source of nitrogen [7]. Over the past 60 years, the use of nitrogen fertilizer per unit of arable land has increased approximately eightfold in order to increase food production, resulting in significant soil, water, and air pollution [8]. Therefore, it is critically important to study the molecular mechanisms of crop nitrogen uptake and utilization and to improve crop nitrogen use efficiency to protect food and environmental security. Meanwhile, nitrate is also an important signaling molecule for lateral root development, flowering, and synergistic absorption of other nutrients [9]. Crucially, this nitrate signaling is perceived and transduced by NODULE INCEPTION-like proteins (NIN-like proteins/NLPs), which serve as pivotal transcription factors and central regulators. NLPs coordinate transcription, transport, metabolism, and systemic growth programs in response to nitrate [10,11,12,13,14,15,16].

Numerous studies have demonstrated that NLPs are central regulators of nitrate signaling and nitrogen uptake and utilization. The number of NLPs varied among different species, including 9 in Arabidopsis, 6 in rice, 9 in maize, 6 in tomato, 33 in tea plant, 18 in wheat, 31 in Brassica, 5 in apple, and 7 in Foxtail Millet [5,10,17,18,19,20,21,22,23]. Despite this diversity across the plant kingdom, NLPs were first identified not in these model crops but in the legume *Lotus japonicus*, where they regulate symbiotic root nodule formation [24]. In Arabidopsis, AtNLP6/7 were phosphorylated by AtCPK10/30/32 to ensure their localization in the nucleus for transcriptional activation of the primary nitrate response genes [12]. Furthermore, NLP6/7 interacts with TCP transcription factors to promote root meristem elongation under nitrogen starvation [15]. Crucially, NLP7 functions as a primary nitrate sensor in plants and plays an important role in the absorption and utilization of nitrogen [4]. NLP8 was reported as a key regulator of nitrate-promoted seed germination in Arabidopsis [25]. In rice, OsNLP3/4 regulates nitrogen nutrient signaling to promote panicle development [26], and OsNLP4 plays a pivotal role in rice nitrogen use efficiency (NUE) and provides insight into crop NUE improvement [27]. In maize, *ZmNLP6* and *ZmNLP8* restore nitrate signaling and assimilation [28]. In *Medicago truncatula*, NLP1 is required for the expression of nitrate-responsive genes [29]. Additionally, NLP1 was reported to regulate *NRT2.1* to mediate root nodule formation across nitrate concentrations [30]. However, there is no advanced research about NLPs in sweet potato.

Nitrogen is vital for plant development, yet both deficiency and excess application compromise sweet potato yields. While nitrogen supply influences carbon metabolism in storage organs, few investigations have explored its impact on non-structural carbohydrate accumulation during sweet potato storage root development. Therefore, it is pivotal to explore the key genes regulating nitrogen uptake and utilization in sweet potato to improve nitrogen use efficiency and increase yield.

Sweet potato (*Ipomoea batatas* (L.) Lam., 2n = 6x = 90) is an economically important root crop that is widely used as an industrial and bioenergy resource worldwide [31]. It possesses a sizable and intricate genome, resulting from polyploidy and high heterozygosity. In recent years, genome assemblies of the hexaploid cultivar Taizhong 6 [32], along with two closely related diploid relatives—*Ipomoea trifida* NCNSP0306 (2n = 2x = 30) and *Ipomoea triloba* NCNSP0323 (2n = 2x = 30) [33]—have been published. The assemblies of hexaploid and diploid genomes make it possible to identify and analyze important gene families at the whole-genome level in sweet potato.

In this study, a total of 28 NLPs were identified in *I. batatas* (7), *Ipomoea trifida* (9), and *Ipomoea triloba* (12). They were classified into four subgroups. We systematically investigated the protein physicochemical properties, chromosome localization, syntenic analysis maps, phylogenetic relationships, gene structure, promoter *cis*-elements, and protein interaction networks of NLPs in sweet potato. Furthermore, tissue specificity and expression pattern analyses of tuberous root development in different accessions and nitrate induction of NLPs were analyzed using qRT-PCR and RNA-seq. Differences in evolution and functions related to development, nitrate signaling response, and abiotic stress response were identified between sweet potato and its two diploid relatives.

## 2. Results

### 2.1. Identification and Characteristics of NLPs in Sweet Potato and Its Two Diploid Relatives

To comprehensively identify all *NLPs* in sweet potato and its two diploid relatives, we employed three typical strategies, as follows: BLASTP, HMMER search, and CD-Search database screening. A total of 28 *NLPs* were identified in *I. batatas* (7), *I. trifida* (9), and *I. triloba* (12) (Table 1 and Appendix A), which were named after “*Ib*”, “*Itf*”, and “*Itb*”, respectively. The physicochemical properties were analyzed using the sequence of *IbNLPs* (Table 1). The CDS length varied from 714 bp (IbNLP1) to 2823 bp (IbNLP7). The amino acid lengths of IbNLPs ranged from 237 aa (IbNLP1) to 940 aa (IbNLP7), with molecular weight (MW) varying from 36.67 kDa (IbNLP1) to 109.26 kDa (IbNLP4). The pI of other NLPs ranged from 5.26 (IbNLP5) to 8.43 (*IbNLP1*), suggesting that they were basic proteins. All IbNLPs contained Ser, Thr, and Tyr phosphorylation sites. The grand average of hydropathicity (GRAVY) value of all IbNLPs proteins varied from −0.67 (IbNLP1) to −0.36 (IbNLP4), indicating that they were hydrophobic. Subcellular localization prediction assays showed that most of the IbNLPs were located in the nucleus, except for IbNLP2, which was located in the nucleus and mitochondrion.

### 2.2. Chromosomal Distribution and Syntenic Analysis of IbNLP Genes

The *NLPs* were distributed across 5, 5, and 6 chromosomes of *I. batatas*, *I. trifida*, and *I. triloba* (Figure 1). In *I. batatas*, two *IbNLPs* were detected on LG2 and LG9, and one each on LG1, LG5, and LG14. In *I. trifida*, three *ItfNLPs* were detected on Chr10, two on Chr01 and Chr15, and one each on Chr09 and Chr12. In *I. triloba*, five *ItbNLPs* were detected on Chr04, three on Chr10, and one each on Chr05, Chr09, Chr12, and Chr15.

Furthermore, four syntenic analysis maps—between *I. batatas* and each of its two diploid relatives, *Arabidopsis*, *Zea mays*, and *Oryza sativa*—were constructed to analyze the phylogenetic mechanisms of *IbNLPs* (Figure 2). Cultivated hexaploid sweet potato *IbNLPs* showed 7 syntenic gene pairs with *I. triloba*, 4 with *I. trifida*, 7 with *Arabidopsis*, 3 with *Zea mays*, and 0 with *Oryza sativa.* Most background collinear blocks associated with *NLP* pairs, identified between *I. batatas* and its two diploid relatives/the dicotyledon *Arabidopsis*, contained more genes than those between *I. batatas* and the monocotyledon rice/maize. *NLPs* showed no collinearity between sweet potato and rice, indicating different paths in their evolutionary history. *IbNLP6* and *IbNLP7* were found in the other three comparative syntenic maps, suggesting that these orthologous pairs might have already existed before the evolutionary divergence of monocotyledons and dicotyledons. In addition, these two genes might have played fundamental roles in the *NLP* family.

### 2.3. Phylogenetic Relationships of NLPs in Sweet Potato and Its Two Diploid Relatives

To study the evolutionary relationships of NLPs in *I. batatas*, *I. trifida*, *I. triloba*, and *Arabidopsis*, we constructed a phylogenetic tree for 37 NLPs from these four species (i.e., 7 in *I. batatas*, 9 in *I. trifida*, 12 in *I. triloba*, and 9 in *Arabidopsis*) (Figure 3). All NLPs were unevenly distributed on each branch of the phylogenetic tree. Interestingly, NLPs in *I. batatas*, *I. trifida*, and *I. triloba* were divided into four subgroups (Group I to IV, Figure 3), but those in *Arabidopsis* were divided into three subgroups (Group I to III). The specific distribution of NLPs was as follows (total: *I. batatas*, *I. trifida*, *I. triloba*, *Arabidopsis*): Group I (13:2,3,3,5), Group II (7:1,2,2,2), Group III (5:1,1,1,2), and Group IV (12:3,6,3,0) (Figure 3; Appendix A). We named IbNLPs, ItfNLPs, and ItbNLPs based on the order in which they appeared on the chromosomes. Only AtNLP4/5/6/9 from *Arabidopsis* had homologous proteins in *I. batatas*, *I. trifida*, and *I. triloba*. These results indicate that the number and types of NLPs distributed in each subgroup of sweet potato differed from those in its two diploid relatives and *Arabidopsis*.

### 2.4. Conserved Domain and Exon–Intron Structure Analysis of NLPs in Sweet Potato and Its Two Diploid Relatives

Protein domains and gene structure are important in analyzing gene functions. We analyzed conserved domains in the 7 IbNLPs, 9 ItfNLPs, and 12 ItbNLPs (Figure 4A). All NLPs contain two conserved domains: an RWP-RK domain, which can bind nitrate-response elements to regulate gene expression, and a PB1 domain, which determines protein interactions. Especially, in Group IV, IbNLP2, ItfNLP2, ItbNLP4, and ItbNLP1 contain 2 RWP-RK domains. ItbNLP1 contains 2 PB1 domains. Notably, IbNLP1 and ItbNLP6 contain another NAM domain, suggesting that they may possess some functions of the NAC transcription factor.

To better understand the structural diversity among *NLPs*, the exon–intron structures were analyzed (Figure 4B). The number of exons in the *NLPs* ranged from two to eight. In more detail, the *NLPs* of Group I contained four to seven exons; the *NLPs* of Group II contained five or six exons; the *NLPs* of Group III contained five or eight exons, and the *NLPs* of Group IV contained two to six exons (Figure 4B). The exon–intron structures of some homologous *NLPs* were different in *I. batatas* compared to those in *I. trifida* and *I. triloba*, such as *IbNLP5* (containing 8 exons), *ItfNLP9* (containing five exons), and *ItbNLP9* (containing five exons) in Group III, and *IbNLP4* (containing five exons), *ItfNLP7* (containing six exons) and *ItbNLP11* (containing five exons) in Group II (Figure 4B). These results indicated that the *NLP* family may have undergone a lineage-specific differentiation event in the sweet potato genome.

### 2.5. cis-Element Analysis in the Promoter of IbNLPs in Sweet Potato

*NLPs* are involved in light, development, and abiotic/biotic response, and *cis*-elements in the promoter region play a key role in the expression of *NLPs*. We analyzed the 2000 bp upstream promoter sequences of *NLPs* from *I. batatas* to predict gene function. We divided the elements into four categories: core, light-responsive elements, development regulation, and abiotic/biotic stress-responsive (Figure 5). A large number of core elements were identified in the 7 *IbNLPs* (CAAT-box and TATA-box) (Figure 5). Most of the *IbNLPs* contained light-responsive elements such as the GT1 motif, which is believed to affect the growth and development of plants (found in *IbNLP1*, *IbNLP2*, *IbNLP3*, *IbNLP4*, *IbNLP5*, and *IbNLP7*). All *IbNLPs* contained many development elements, such as the AT-rich element, which plays a key role in the structure and function of the eukaryotic genome (found in *IbNLP1* and *IbNLP2*), CAT-box, which is involved in meristem formation (found in *IbNLP4*, *IbNLP5*, and *IbNLP6*), and the circadian element, which is the core regulator of the circadian clock (found in *IbNLP2* and *IbNLP3*).

Crucially, all *IbNLPs* contained drought-responsive elements MYB and MYC elements, which indicated that they may be involved in the drought response (Figure 5). Meanwhile, *IbNLP1, IbNLP2*, and *IbNLP3* contained the salt-responsive element MBS and W box. Furthermore, Most NLPs contained AAGAA-motif, ARE, ERE, and WUN-motif elements, which are involved in abiotic/biotic stress-response. In summary, *IbNLPs* are involved in light response, plant growth and development, and abiotic stress adaptation in sweet potato.

### 2.6. Protein Interaction Network of IbNLPs in Sweet Potato

To explore the potential regulatory network of IbNLPs, we constructed an IbNLPs interaction network based on *Arabidopsis* orthologous proteins (Figure 6). Protein interaction predictions indicated that some IbNLPs (IbNLP4, IbNLP5, IbNLP6, and IbNLP7) could interact with each other to form a heterodimer. In addition, NLPs could interact with the transcription factor HRS1, which can integrate nitrate and phosphate starvation responses and adaptation of root architecture depending on nutrient availability [34]. NLPs could interact with MOB1A/B, which play critical roles in plant development [35,36]. NLPs could interact with ROPGEF14, which is involved in the response to complex environmental signals [37]. These results indicated that IbNLPs were involved in the regulation of nitrate signaling response, plant growth and development, and stress adaptation in sweet potato.

### 2.7. Expression Analysis of NLPs in Sweet Potato and Its Two Diploid Relatives

#### 2.7.1. Expression Analysis in Different Tissues

To investigate the potential biological function of *IbNLPs* in growth and development, we measured the expression of *IbNLPs* in six tissues (i.e., bud, petiole, leaf, stem, fibrous root, and tuberous root) (Figure 7A). In Group I, *IbNLP6* was highly expressed in the fibrous root, indicating that *IbNLP6* may be involved in nitrogen uptake in the fibrous root. Meanwhile, *IbNLP7* and *IbNLP4* were highly expressed in the bud and stem. *IbNLP5* in Group III showed higher expression levels in all tissues than in other subgroups. In Group IV, *IbNLP1*, *IbNLP2*, and *IbNLP3* were lowly expressed in all tissues.

In addition, we used RNA-seq data of six tissues (i.e., flower bud, flower, leaf, stem, root1, and root2) to study the expression patterns of *NLPs* in *I. trifida* and *I. triloba* [33] (Figure 7B,C). In *I. trifida*, *ItfNLP9* in Group III was highly expressed in six tissues. *ItfNLP3* was highly expressed in the flower bud, root1, and root2. *ItfNLP6* was highly expressed in the flower and leaf (Figure 7B). *ItfNLPs* in Group IV were lowly expressed in all tissues. In *I. triloba*, *ItbNLP10* and *ItbNLP11* were highly expressed in the flower bud and leaf; *ItbNLP2*, *ItbNLP9*, and *ItbNLP10* were highly expressed in the flower; *ItbNLP5* and *ItbNLP11* were highly expressed in the stem (Figure 7C). Therefore, we observed that *ItbNLP2*, *ItbNLP7*, and *ItbNLP12* were highly expressed in the root1 and root2, which suggested that they may regulate root development by regulating nutrient uptake (Figure 7C). Those results showed that *NLPs* exhibit different expression patterns and play important roles in the growth and development of sweet potato and the two diploids.

#### 2.7.2. Expression Analysis in Different Development Stages

We further performed qRT-PCR to evaluate the expression levels of *IbNLPs* in different development stages of sweet potato roots (i.e., 10 d, 20 d, 30 d, 40 d, 50 d, 60 d, 70 d, 80 d, 100 d, and 120 d) (Figure 8). All *IbNLPs* were highly expressed in the early stages of sweet potato root development (i.e., 10 d), suggesting that they may be involved in the initial expansion of sweet potato roots. *IbNLP6* was extremely highly expressed in the early stage of root development, and its expression reached 33-fold at 10 d. *IbNLP7* was highly expressed in the late stage of root development (i.e., 80 d, 100 d). In Group II, *IbNLP4* was expressed highly with 2.75-fold in the early stage. Especially, *IbNLP5* showed a high expression level in all stages of root development, notably, 80 d and 100 d of tuberous root development. Interestingly, *IbNLP1*, *IbNLP2*, and *IbNLP3* in Group IV were all expressed less than 0.5-fold in all stages.

#### 2.7.3. Expression Analysis in Sweet Potato Starch Transcriptome

As a root crop, the most important product of the sweet potato is starch. We analyzed the expression of *IbNLPs* using two published starch-related transcriptomes (Figure 9) [38,39]. The expression level of *IbNLP2*, *IbNLP5*, and *IbNLP7* in the four high-starch accessions (H8, H21, H42, H74) was higher than that in the four low-starch accessions (L4, L18, L118, L123) (Figure 9A). Moreover, the expression of *IbNLP5* in high-starch cultivar H283 was higher than in low-starch cultivar L423 at three different root development stages in tuberous roots (Figure 9B). Notably, all *NLPs* showed high expression levels in high starch, except *IbNLP4* and *IbNLP3* (Figure 9A). However, in Group IV, all *NLPs* were highly expressed at 60 d in L423, which was a low-starch accession (Figure 9B). Those results showed that *NLPs* exhibit different expression patterns, and *IbNLP5* may play an important role in starch synthesis in sweet potato.

#### 2.7.4. Expression Analysis in Nitrate Induction

Nitrogen is a limiting factor for plant growth and has an important impact on sweet potato yields [1,40,41]. *NLPs* are the key transcription factors in nitrate signaling to regulate nitrogen uptake and utilization in crops. To obtain evidence of possible roles of *IbNLPs* in root nitrate absorption regulation during nitrogen deficiency, the transcript abundance of *IbNLPs* in roots was examined by qRT-PCR after N starvation (Figure 10). We found that all *NLPs* responded quickly to nitrogen-induced expression at 0.5 h after treatment. In particular, we noted that *IbNLP6* was up-regulated by up to 29-fold and *IbNLP4* was up-regulated by up to 15-fold, which suggests that they may be key regulators of the nitrogen signaling pathway. Notably, some *IbNLPs* were also induced at other times, such as *IbNLP6* (1 h), *IbNLP4* (3 h and 6 h), *IbNLP3*, and *IbNLP2* (24 h). In summary, *NLPs* are involved in nitrate signaling response and nitrogen uptake and utilization in sweet potato.

#### 2.7.5. Expression Analysis Under Abiotic Stresses

A large number of stress-related elements were found in the promoter regions of *NLPs*, indicating that they could participate in the stress-related processes in sweet potato. We analyzed the expression patterns of *IbNLPs* using the RNA-seq data of a drought-tolerant variety Xu55-2 under drought stress, and the RNA-seq data of a salt-sensitive variety Lizixiang and a salt-tolerant line ND98 under salt stress [42,43]. *IbNLP4* was induced by both PEG and NaCl treatments in Xu55-2, Lizixiang, and ND98 (Figure 11). In Group I, *IbNLP6* and *IbNLP7* were inhibited by PEG treatments in Xu55-2. In Group IV, *IbNLP3* was up-regulated to 174-fold by PEG at 48 h in Xu55-2, and was induced 5-fold in Lizixiang by NaCl treatment. *IbNLP1* and *IbNLP5* showed no significant change in expression levels in Xu55-2, Lizixiang, and ND98. *IbNLP2* and *IbNLP7* were slightly inhibited by PEG in Xu55-2.

Further, we also analyzed the expression patterns of *NLPs* using the RNA-seq data of *I. trifida* and *I. triloba* under drought and salt treatments [33]. In *I. trifida*, *ItfNLP8* and *ItfNLP9* were induced by both drought and salt treatments, which indicated that they were involved in the response to drought stress. *ItfNLP4*, *ItfNLP6*, and *ItfNLP7* were inhibited by both drought and salt treatments, which suggested that they may be involved in life activities inhibited by drought stress, such as nitrogen uptake and nitrate signaling response (Appendix A). In *I. triloba*, *ItbNLP9*, *ItbNLP3*, and *ItbNLP2* were induced by both drought and salt treatments; *ItbNLP8*, *ItfNLP10*, and *ItfNLP11* were inhibited by both drought and salt treatments (Appendix A). Taken together, these results indicated that *NLPs* were differentially expressed in response to various abiotic stresses between sweet potato and its two diploid relatives.

## 3. Discussion

Nitrogen (N) is an essential nutrient for photosynthetic plant growth in most soils and controls metabolic and developmental processes pivotal to plant vegetative and reproductive development [1,2,3,4,6,7,8,9]. NLPs are plant-specific transcription factors that play crucial roles in nitrate signaling response [5,10,11,12]. However, the functions and transcriptional regulatory mechanisms of NLPs remain largely unknown in sweet potato. The release of the sweet potato genome provides the possibility for genome-wide identification of NLPs. In this study, we systematically identified NLPs and compared their characteristics between the cultivated hexaploid sweet potato and its two diploid relatives based on their genome sequences. Our study provides insights into the function of NLPs in sweet potato.

### 3.1. Identification and Evolution of the NLP Family

In this study, a total of 28 NLPs were identified in sweet potato and its two diploid relatives. The number of NLPs was identified as 7 in *I. batatas*, 9 in *I. trifida*, and 12 in *I. triloba.* (Table 1 and Appendix A). Fewer *NLPs* in hexaploid sweet potato than diploid relatives, suggesting that some *NLPs* (i.e., *ItfNLP6* and *ItfNLP8* in *I. trifida*; *ItbNLP1*, *ItbNLP2*, *ItbNLP4*, *ItbNLP10*, and *ItbNLP12* in *I. triloba*) were lost during domestication. There may be functional redundancy between *NLPs*, such as in Group I, where *ItfNLP8 and ItfNLP3* showed similar expression patterns in different tissues of *I. trifida* (Figure 7B). In *I. triloba*, *ItbNLP1* and *ItbNLP2* showed low expression (less than 0.3-fold) in all tissues, suggesting that they may not be functional (Figure 7C). The chromosome localization and distribution of NLPs were different between *I. batatas*, *I. trifida*, and *I. triloba*; 6 chromosomes contained NLP genes in *I. batatas* and *I. triloba*, but only 5 in *I. trifida* (Figure 1). Moreover, collinearity analysis revealed that there are 7 syntenic gene pairs with *I. triloba*, but 4 with *I. trifida* (Figure 2A). Genomic alignment revealed differentiation and evolution of chromosomes. Based on the phylogenetic relationship, NLPs in *I. batatas*, *I. trifida*, and *I. triloba* were divided into four subgroups (Groups I to IV, Figure 3), but AtNLPs were divided into three subgroups (Groups I to III). The number and types of NLPs distributed in each subgroup of sweet potato and its two diploid relatives were different from those of Arabidopsis and other plants. Moreover, only AtNLP4/5/6/9 from *Arabidopsis* have homologous proteins in *I. batatas*, *I. trifida*, and *I. triloba* (Table 1; Figure 3). These results suggest that the NLP family might have undergone a lineage-specific differentiation event in the terrestrial plant genome.

All NLPs contain two conserved domains, an RWP-RK domain and a PB1 domain (Figure 4A). The RWP-RK domain recognizes nitrate-responsive *cis*-elements (NREs), thereby mediating DNA binding and regulating gene expression in midstream and downstream nitrogen pathways. The PB1 domain determines protein interactions [17]. The two domains are key to NLPs’ roles in plant growth and development, stress response, and nitrogen regulation [13,14,15,16]. Introns typically function as mutation-resistant buffer zones, minimizing deleterious mutations and insertions. Furthermore, they play essential roles in regulating mRNA export, facilitating transcriptional coupling, enabling alternative splicing, modulating gene expression, and other fundamental cellular processes. The exon–intron structures of some homologous *NLPs* were different in *I. batatas* compared to those in *I. trifida* and *I. triloba*, such as *IbNLP5* (containing 8 exons), *ItfNLP9* (containing five exons), and *ItbNLP9* (containing five exons) in Group III, *IbNLP4* (containing five exons), *ItfNLP7* (containing six exons) and *ItbNLP11* (containing five exons) in Group II (Figure 4B). These differences in the exon–intron structures in sweet potato and its two diploid relatives might lead *NLPs* to be involved in different growth and development processes in sweet potato.

### 3.2. Different Functions of NLPs in Tuberous Root Development and Starch Synthesis in Sweet Potato

Nitrate acts as a signal and nutrient to regulate root growth and development [44,45]. In Arabidopsis, NLP6/7 interact with TCP20 to promote root meristem elongation under nitrogen starvation [15]; furthermore, overexpression of AtNLP7 promotes primary and lateral root growth [46]. In rice, the root length of *nlp1* mutants is shorter than that of the wild type, and overexpression of *OsNLP1* significantly increases root length [47]. As the key regulators in this process in plants, NLPs play vital roles in the tuberous root development of sweet potato. In this study, in the N starvation experiment, all NLPs responded quickly to nitrogen-induced expression at 0.5 h after treatment in sweet potato. In particular, we noted that *IbNLP6*, which is conserved in different plants (Figure 4), was up-regulated by up to 29-fold, which suggests that it may be a key regulator of the nitrogen signaling pathway.

For sweet potato, the formation and development of tuberous roots are critical to its yield and quality. Meanwhile, the development of the root system and nitrogen uptake and utilization directly affect the yield of tuberous roots [40,41]. Most of the *IbNLPs* were highly expressed in the early stage of root development. Notably, we found that *IbNLP6*, which is highly induced by N, showed the highest expression level in the early stage of root development. This result suggests that it may be involved in the uptake and utilization of nitrogen in sweet potato fibrous roots.

As a food crop, the most important component of sweet potato is starch. The expression levels of *IbNLP2*, *IbNLP5*, and *IbNLP7* in the four high-starch accessions (H8, H21, H42, H74) were higher than those in the four low-starch accessions (L4, L18, L118, L123) (Figure 9A) [38]. Moreover, the expression of *IbNLP5* in high-starch cultivar H283 was higher than in low-starch cultivar L423 at three different root development stages in tuberous roots (Figure 9B) [39]. These results suggest that *NLPs* may play an important role in starch synthesis in sweet potato.

### 3.3. Different Functions of NLPs on Abiotic Stress Response Between Sweet Potato and Its Two Diploid Relatives

The *NLPs* have been reported to participate in the abiotic stress response in plants. *AtNLP7* in Arabidopsis may be involved in stomatal movement and drought resistance [11]. The *NLPs* of *Poncirus Trifoliata* were expressed differently under different water conditions. *NLPs* in roots continued to downregulate under drought stress [48]. Under drought stress, the expression levels of *MdNLP2*, *MdNLP3*, and *MdNLP5* in apple showed a trend of first increasing and then decreasing [49]. In this study, we found *NLPs* were differentially expressed in response to various abiotic stresses between sweet potato and its two diploid relatives. *IbNLP4* was induced by both PEG and NaCl treatments in Xu55-2, Lizixiang, and ND98 (Figure 11). *IbNLP6* was inhibited by PEG and NaCl treatments in Xu55-2 and Lizixiang. Moreover, diploid *I. trifida* and *I. triloba* represent valuable resources for identifying functional genes—particularly those conferring resistance or tolerance to biotic and abiotic stresses—that may have been lost during sweet potato domestication. *ItfNLP8*, *ItfNLP9*, *ItbNLP9*, *ItbNLP3*, and *ItbNLP2* were induced by both drought and salt treatments, suggesting that they are involved in the response to drought stress. Taken together, these results indicate that *NLP*s can be candidate genes to improve abiotic stress tolerance in sweet potato.

## 4. Materials and Methods

### 4.1. Identification of NLPs

The whole genome sequences of *I. batatas*, *I. trifida*, and *I. triloba* were downloaded from Ipomoea Genome Hub (https://ipomoea-genome.org/ (accessed on 15 June 2025)) and Sweetpotato Genomics Resource (http://sweetpotato.plantbiology.msu.edu/ (accessed on 15 June 2025)). To accurately identify all NLP family members, three different screening methods were combined. First, the BLAST (https://www.arabidopsis.org/ (accessed on 15 June 2025)) algorithm was used to identify predicted NLPs using all AtNLPs from the Arabidopsis genome database (https://www.arabidopsis.org/ (accessed on 15 June 2025)) as queries (BLASTP, E value ≤ 1 × 10^−5^). Next, the HMMER 3.0 software was used to identify potential NLPs through the Hidden Markov Model profiles (HMMER search, E value ≤ 1 × 10^−5^) of PF02042 and PB1 domains (hmm, PF00564, which were extracted from the Pfam databases (http://pfam.xfam.org/ (accessed on 15 June 2025)). Finally, all putative NLPs were verified using SMART (http://smart.embl-heidelberg.de/ (accessed on 15 June 2025)) and CD-Search (https://www.ncbi.nlm.nih.gov/Structure/cdd/wrpsb.cgi (accessed on 15 June 2025)).

### 4.2. Chromosomal Distribution of NLPs and Syntenic Analysis

The *IbNLPs*, *ItfNLPs*, and *ItbNLPs* were separately mapped to the *I. batatas*, *I. trifida*, and *I. triloba* chromosomes based on the chromosomal location provided in the Ipomoea Genome Hub (https://ipomoea-genome.org/ (accessed on 16 June 2025)) and Sweetpotato Genomics Resource (http://sweetpotato.plantbiology.msu.edu/ (accessed on 16 June 2025)). The visualization was generated using the TBtools-II software (v.2.315) (South China Agricultural University, Guangzhou, China) [50]. The syntenic analysis maps of orthologous NLP genes were constructed using the Dual Synteny Plotter software (v2.0)) [50].

### 4.3. Protein Properties Prediction of NLPs

The phosphorylation sites of NLPs were predicted using GPS 5.0 [51]. The MW, theoretical pI, and hydropathy of NLPs were calculated with ExPASy (https://www.expasy.org/ (accessed on 18 June 2025)). The subcellular localization of NLPs was predicted using Plant-mPLoc (http://www.csbio.sjtu.edu.cn/bioinf/plant-multi/ (accessed on 18 June 2025)).

### 4.4. Phylogenetic Analysis of NLPs

The phylogenetic analysis of NLPs from *I. batatas*, *I. trifida*, *I. triloba*, and *Arabidopsis* was performed using ClustalW (v2.0.11) in MEGA 7.0 [52] with default parameters. The maximum likelihood method and the Poisson correction model were used. Bootstrapping was performed 1000 times. Then, the phylogenetic tree was constructed using iTOL (http://itol.embl.de/ (accessed on 16 June 2025) v7.2.1).

### 4.5. Domain Identification Analysis of NLPs

The conserved domains of NLPs were analyzed using NCBI (https://www.ncbi.nlm.nih.gov/Structure/cdd/wrpsb.cgi (accessed on 16 June 2025) (v2.1)).

### 4.6. Exon–Intron Structures and Promoter Analysis of NLPs

The exon–intron structures of *NLPs* were obtained using GSDS 2.0 (http://gsds.gao-lab.org/) and were visualized using the TBtools-II software. *cis*-elements in the approximately 2000 bp promoter regions of *NLPs* were predicted using PlantCARE (http://bioinformatics.psb.ugent.be/webtools/plantcare/html/ (accessed on 18 June 2025)) [53].

### 4.7. Protein Interaction Network of NLPs

Protein interaction networks of NLPs were predicted using STRING (https://cn.string-db.org/ (v12.0)), based on Arabidopsis homologous proteins. The network map was built using Cytoscape software (v3.0) [54].

### 4.8. qRT-PCR Analysis of NLPs

The sweet potato (*I. batatas*) variety ‘Xushu18’ was used for qRT-PCR analysis in this study. In vitro-grown Xushu18 plants were cultured on Murashige and Skoog (MS) medium at 27 ± 1 °C under a photoperiod consisting of 13 h of cool-white fluorescent light at 54 μmol m^−2^ s^−1^ and 11 h of darkness. Sweet potato plants were cultivated in a field on the campus of China Agricultural University, Beijing, China. For expression analysis in various tissues, total RNA was extracted from the bud, leaves, petioles, stems, fibrous root, and tuberous root tissues of 3-month-old field-grown Xushu18 plants; different tuberous root tissues of Xushu18 at different developmental stages (10 d, 20 d, 30 d, 40 d, 50 d, 60 d, 80 d, 100 d, 120 d) were collected using the TRIzol method (Invitrogen, Carlsbad, CA, USA). For expression analysis of nitrate response, 1-month-old in vitro Xushu18 seedlings were subjected to N starvation (0.15 mM KNO_3_) for 2 weeks and then supplied with nitrate (5 mM KNO_3_). A modified Hoagland nutrient solution was employed, with 5 mM KNO_3_ as a sufficient nitrogen solution and 0.15 mM KNO_3_ as an N starvation solution. The differences in potassium supply were balanced with KCl. The solutions were changed every 2 days. The roots were sampled at 0, 0.5, 1, 3, 6, 12, and 24 h after being treated. Three independent biological replicates were taken, each with three plants. qRT-PCR was conducted using the SYBR detection protocol (TaKaRa, Kyoto, Japan) on a 7500 Real-Time PCR system (Applied Biosystems, Foster City, CA, USA). The reaction mixture was composed of first-strand cDNA, primer mix, and SYBR Green M Mix (TaKaRa; code RR420A) in a final volume of 20 μL. The sweet potato actin gene (GenBank AY905538) was used as the internal control. The relative gene expression levels were quantified using the comparative C_T_ method [55]. The specific primers used for qRT-PCR analysis are listed in Appendix A. The heat maps of gene expression profiles were constructed using TBtools-II software (v.2.315) [50].

### 4.9. Transcriptome Analysis

The RNA-seq data of *ItfNLPs* and *ItbNLPs* in *I. trifida* and *I. triloba* were downloaded from the Sweetpotato Genomics Resource (http://sweetpotato.plantbiology.msu.edu/ (accessed on 18 June 2025)). The RNA-seq data of *IbNLPs* in *I. batatas* were obtained from related research in our laboratory [44,45]. The expression levels of *NLPs* were calculated as fragments per kilobase of exon per million fragments mapped (FPKM). The heat maps were constructed using TBtools-II software (v.2.315) [50].

## 5. Conclusions

In this study, we systematically investigated the protein physicochemical properties, chromosome localization, phylogenetic relationships, syntenic analysis maps, gene structure, *cis*-elements of promoters, and protein interaction networks of *NLPs* in sweet potato. In addition, tissue specificity and expression pattern analyses of tuberous root development in different accessions and nitrate induction of *NLPs* were analyzed using qRT-PCR and RNA-seq. We discovered the evolution and different functions in development, nitrate signaling response, and abiotic stress response of *NLPs* in sweet potato and its two diploid relatives. In summary, this study is a critical first step in understanding the functions of sweet potato *NLPs* and offers more candidate genes for improving nitrogen use efficiency and increasing yield in cultivated sweet potato.

## Figures and Tables

**Figure 1 ijms-26-08435-f001:**
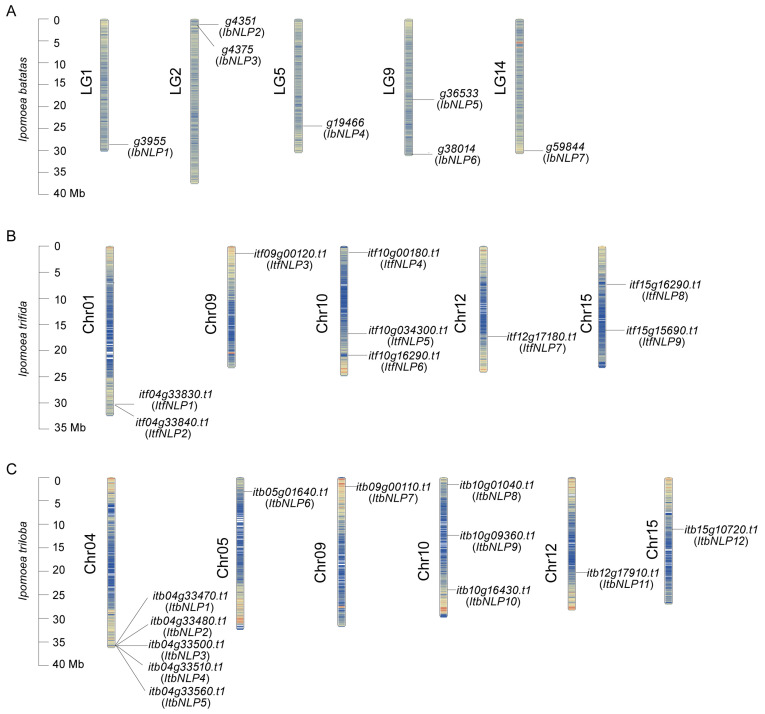
Chromosomal localization and distribution of *NLP*s in *I. batatas* (**A**), *I. trifida* (**B**), and *I. triloba* (**C**). The bars in the left margin represent chromosomes. The chromosome numbers are displayed on the left side of the chromosomes, and the gene names are displayed on the right side. Detailed chromosomal location information is listed in Appendix A.

**Figure 2 ijms-26-08435-f002:**
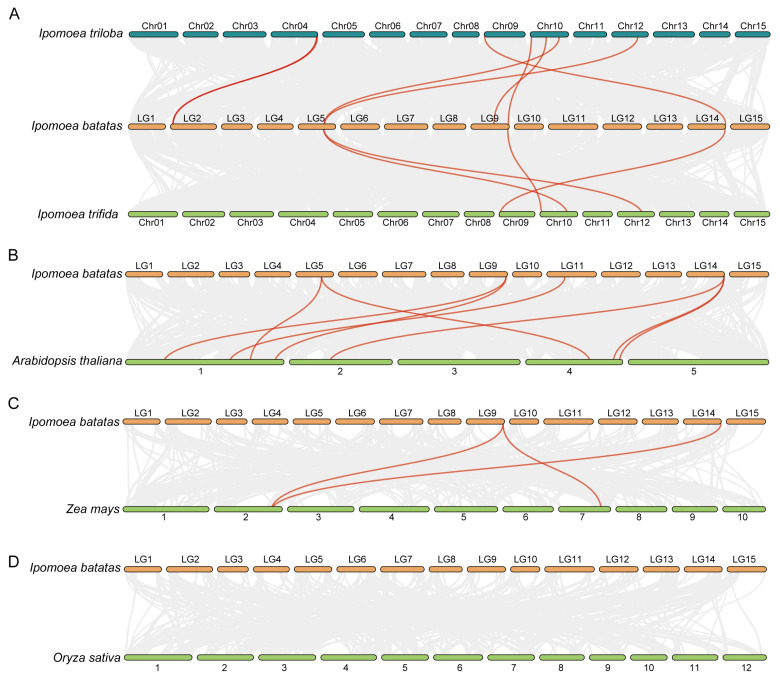
Syntenic *NLP* gene pairs between *I. batatas*, *I. trifida*, and *I. triloba in* (**A**), *I. batatas* and *Arabidopsis* in (**B**), *I. batatas* and *Zea mays* in (**C**), and *I. batatas* and *Oryza sativa* in (**D**). Gray lines indicate all the collinear blocks in the genome, and the red lines indicate the syntenic *NLP* gene pairs.

**Figure 3 ijms-26-08435-f003:**
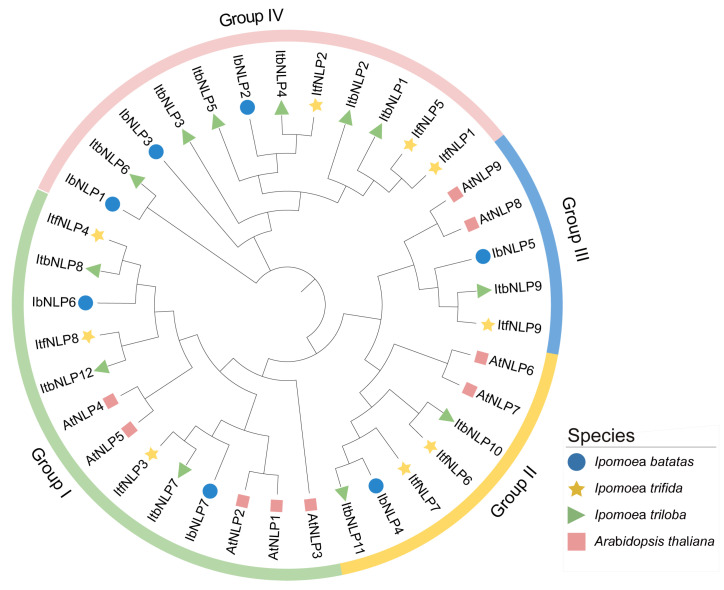
Phylogenetic analysis of the NLPs in *I. batatas*, *I. trifida, I. triloba*, and *Arabidopsis.* A total of 37 NLPs were divided into our subgroups (Groups I to IV) according to the evolutionary distance. The blue circles, yellow pentagrams, green triangles, and pink squares represent IbNLPs in *I. batatas*, ItfNLPs in *I. trifida*, ItbNLPs in *I. triloba*, and AtNLPs in *Arabidopsis*, respectively.

**Figure 4 ijms-26-08435-f004:**
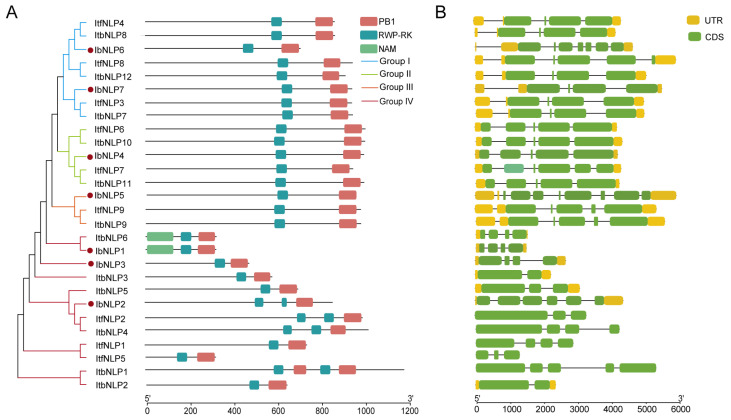
The phylogenetic tree shows that NLPs are distributed into four groups on the left. The red dots represent the IbNLPs. (**A**) Conserved domain structure of NLPs in *I. batatas*, *I. trifida*, and *I. triloba*. The pink, blue, and yellow boxes represent the PB1 domain, RWP-RK domain, and NAM domain, respectively; and (**B**) Exon–intron structure of NLPs in *I. batatas*, *I. trifida*, and *I. triloba*. The yellow boxes, green boxes, and grey lines represent the UTR, exons, and introns, respectively.

**Figure 5 ijms-26-08435-f005:**
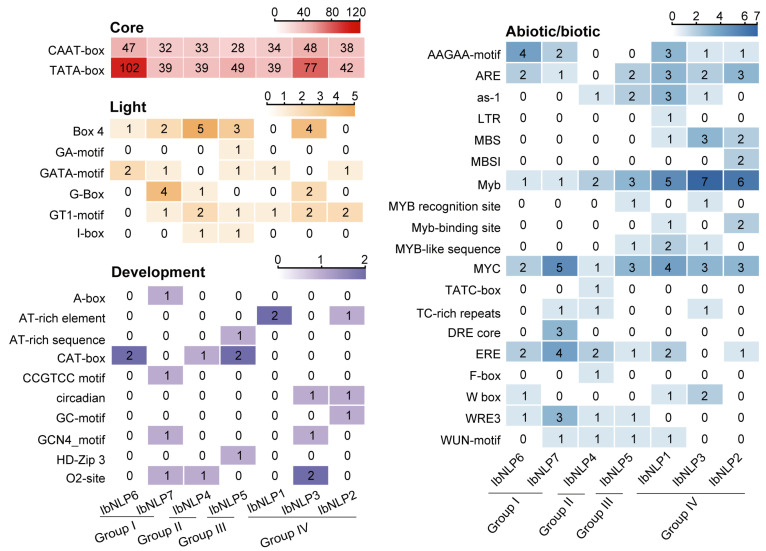
*Cis*-elements analysis of *IbNLPs* in *I. batatas*. The *cis*-elements were divided into four categories. The degree of different colors represents the number of *cis*-elements in the *IbNLPs* promoters.

**Figure 6 ijms-26-08435-f006:**
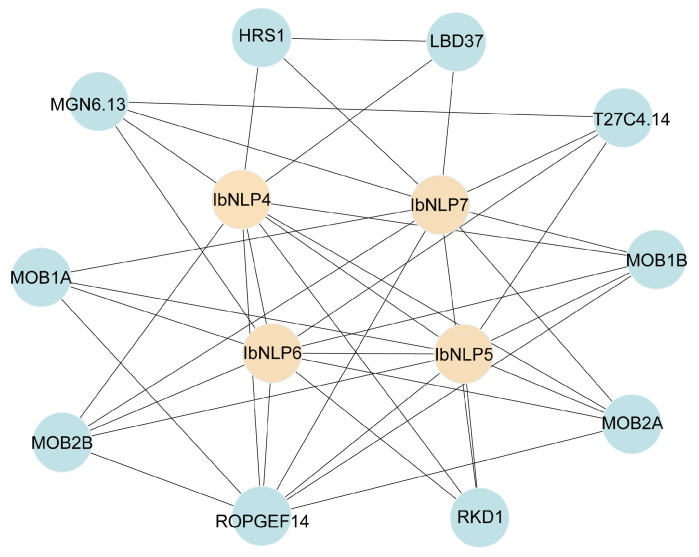
Functional interaction networks of IbNLPs in *I. batatas* according to orthologs in *Arabidopsis*. Network nodes represent proteins, and lines represent protein-protein associations.

**Figure 7 ijms-26-08435-f007:**
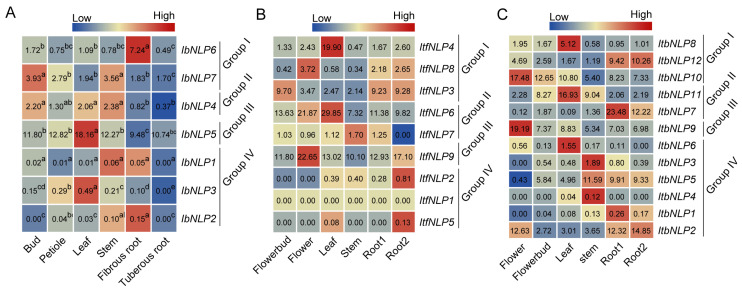
Gene expression patterns of *NLPs* in different tissues of *I. batatas*, *I. trifida*, and *I. triloba*. (**A**) Expression analysis in bud, petiole, leaf, stem, fibrous root, and tuberous root of *I. batatas*. RT-qPCR determined the values from three biological replicates consisting of pools of three plants, and the results were analyzed using the comparative C_T_ method. The expression of *IbNLP6* in the leaf was considered as “1”. The fold change was shown in the boxes. Different lowercase letters indicate a significant difference in each *IbNLP* at *p* < 0.05 based on the Student’s *t*-test. (**B**) Gene expression patterns of *ItfNLPs* in flower bud, flower, leaf, stem, root1, and root2 of *I. trifida* as determined by RNA-seq. Log_2_ (FPKM) was shown in the boxes. (**C**) Gene expression patterns of *ItbNLPs* in flower bud, flower, leaf, stem, root1, and root2 of *I. triloba* as determined by RNA-seq. Log_2_ (FPKM) was shown in the boxes.

**Figure 8 ijms-26-08435-f008:**
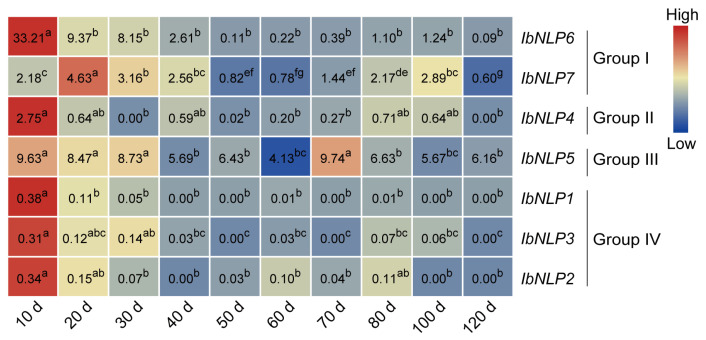
Gene expression patterns of *IbNLPs* in different root development stages (i.e., 10 d, 20 d, 30 d, 40 d, 50 d, 60 d, 70 d, 80 d, 100 d, and 120 d). The values were determined by RT-qPCR from three biological replicates consisting of pools of three plants, and the results were analyzed using the comparative C_T_ method. The expression of *IbNLP6* at 80 d was considered as “1”. The fold changes were shown in the boxes. Different lowercase letters indicate a significant difference for each *IbNLP* at *p* < 0.05 based on Student’s *t*-test.

**Figure 9 ijms-26-08435-f009:**
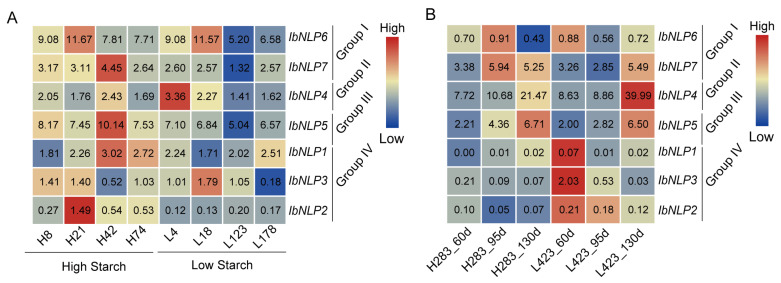
Gene expression of *IbNLPs* in starch-related transcriptomes. (**A**) Expression analysis of *IbNLPs* in high-starch accessions (H8, H21, H42, H74) and low-starch accessions (L4, L18, L118, L123). (**B**) Expression analysis of *IbNLPs* in high-starch accession H283 and low-starch accession L423 at 60, 95, and 130 d after planting. Log_2_ (FPKM) was shown in the boxes.

**Figure 10 ijms-26-08435-f010:**
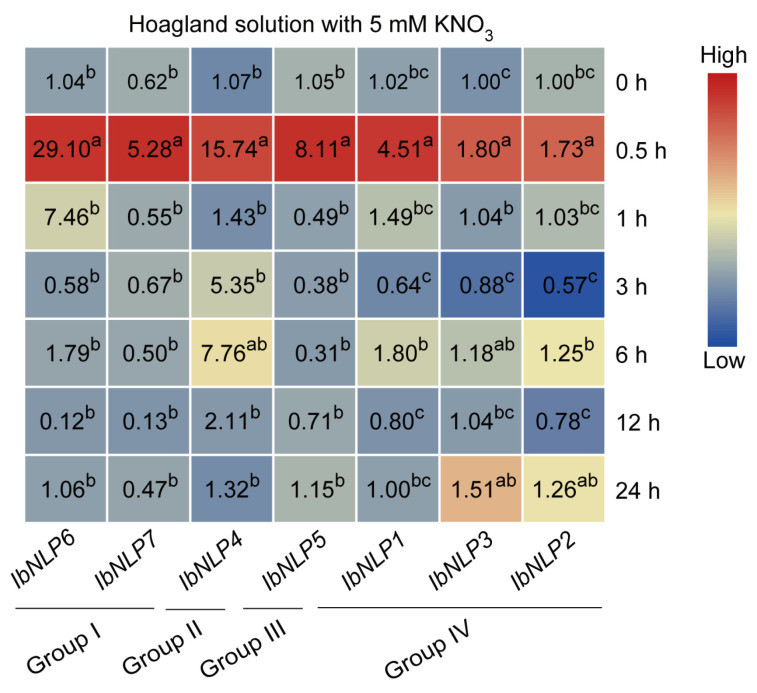
Gene expression of *IbNLPs* after N starvation treatment. Log_2_ (FPKM) was shown in the boxes. The values were determined by RT-qPCR from three biological replicates consisting of pools of three plants, and the results were analyzed using the comparative C_T_ method. The expression of *IbNLP6* at 0 h was considered as “1”. The fold changes were shown in the boxes. Different lowercase letters indicate a significant difference for each *IbNLP* at *p* < 0.05 based on Student’s *t*-test.

**Figure 11 ijms-26-08435-f011:**
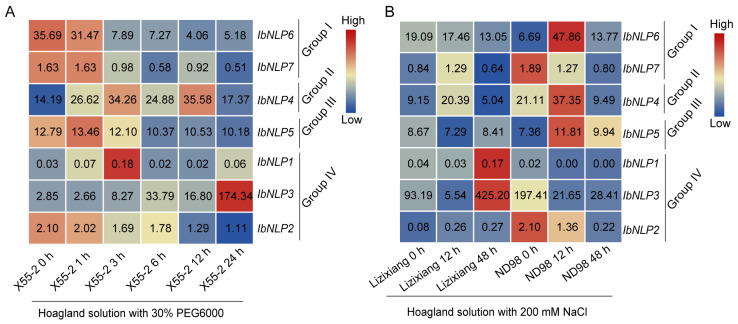
Gene expression of *IbNLPs* under drought and salt stresses as determined by RNA-seq. (**A**) Expression analysis of *IbNLPs* under PEG treatment in a drought-tolerant variety Xu55-2. (**B**) Expression analysis of *IbNLPs* under NaCl treatment in a salt-sensitive variety, Lizixiang, and a salt-tolerant line, ND98. Log2 (FPKM) was shown in the boxes.

**Table 1 ijms-26-08435-t001:** Characterization of IbNLPs in sweet potato.

Gene Name	Gene ID	CDS Length/bp	Protein Size/aa	Phosphorylation Site			MW/kDa	pI	GRAVY	Subcellular Locations	Arabidopsis Homologous
				Ser	Thr	Tyr					
*IbNLP1*	g3955	714	237	2	1	6	36.67	8.43	−0.67	Nucleus	*/*
*IbNLP2*	g4351	1812	603	26	24	10	94.86	5.55	−0.40	Mitochondrion; Nucleus	*/*
*IbNLP3*	g4375	996	331	9	12	8	52.6	6.50	−0.47	Nucleus	*/*
*IbNLP4*	g19466	2088	695	29	17	11	109.26	5.74	−0.36	Nucleus	*AtNLP6*
*IbNLP5*	g36533	2046	681	30	17	8	102.51	5.26	−0.38	Nucleus	*AtNLP9*
*IbNLP6*	g38014	1464	487	15	9	8	79.55	8.30	−0.51	Nucleus	*AtNLP4*
*IbNLP7*	g59844	2823	940	36	27	16	104.09	6.21	−0.57	Nucleus	*AtNLP5*

CDS, coding sequence; MW, molecular weight; pI, isoelectric point.

## Data Availability

The data presented in this study are openly available at reference number [32,33,38,39].

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
