# Peer review of "Genome-Wide Identification and Expression Analysis of the NLP Family in Sweet Potato and Its Two Diploid Relatives"

_ijms, 2025, doi:10.3390/ijms26178435_

Round 1
Reviewer 1 Report
Comments and Suggestions for Authors
The manuscript, entitled "Genome-Wide Identification and Expression Analysis of NLPs Family in Sweet Potato and Its Two Diploid Relatives", publishes the results of a study of the properties of NIN-like proteins and many other related aspects in several types of sweet potatoes. The research was carried out competently, the work seems to be large-scale, deeply thought out and completed. Given that signaling molecules in plants control many vital processes, the study is of scientific interest. The text is written competently, the narrative is consistent, and it is advisable to formulate conclusions more specifically. The graphic design is well done, the number of references to literary sources is satisfactory. It is recommended to accept the article after minor changes. 1) Formal remarks on the design of the article. Line 147. A space is needed between the generic and specific name. Pages 4,5,6,7 and 11. There is too much free space at the end of the page that is not occupied by the text. Line 509. The title is separated from the text of the chapter. It is desirable to shift figures 4, 5, 7, 9 and 11 to the right. The font of the captions to the figures differs from the font of the article text. 2) The article does not decipher the concept of "NIN-like". 3) Lines 515-517. "The evolution, ... were discovered between ...(two species)." It is desirable to formulate this sentence in a different way. 4) The article is of great scientific importance. However, the statement in Chapter 5 about the possible increase in sweet potato yields using the data in this article seems controversial.
Author Response
Reviewers' comments:
Reviewer#1:
The manuscript, entitled "Genome-Wide Identification and Expression Analysis of NLPs Family in Sweet Potato and Its Two Diploid Relatives", publishes the results of a study of the properties of NIN-like proteins and many other related aspects in several types of sweet potatoes. The research was carried out competently, the work seems to be large-scale, deeply thought out and completed. Given that signaling molecules in plants control many vital processes, the study is of scientific interest. The text is written competently, the narrative is consistent, and it is advisable to formulate conclusions more specifically. The graphic design is well done, the number of references to literary sources is satisfactory. It is recommended to accept the article after minor changes.
1) Formal remarks on the design of the article. Line 147. A space is needed between the generic and specific name. Pages 4,5,6,7 and 11. There is too much free space at the end of the page that is not occupied by the text. Line 509. The title is separated from the text of the chapter. It is desirable to shift figures 4, 5, 7, 9 and 11 to the right. The font of the captions to the figures differs from the font of the article text.
Response:
We thank the reviewer for pointing these out. We have revised “I.batatas” to “I. batatas” in line 147. We have adjusted the position of figure 3 in the article to make sure there is no too much free space at the end of the page in pages 4,5,6,7 and11. As suggested, we revised to make sure the title is next to the text of the chapter 5 in line 509. As suggested, we shifted figures 4, 5, 7, 9 and 11 to the right. We checked the font of all chart titles with the font of the article text to make sure they matched.
2) The article does not decipher the concept of "NIN-like".
Response:
We thank the reviewer for pointing this out. As suggested, we deciphered the concept of "NIN-like" in Lines 49-51. “Crucially, this nitrate signaling is perceived and transduced by NODULE INCEPTION-like protein (NIN-like proteins /NLPs), which serve as pivotal transcription factors and central regulators”
3) Lines 515-517. "The evolution, ... were discovered between ...(two species)." It is desirable to formulate this sentence in a different way.
Response:
Thank you for your suggestion. As suggested, we used a different way to describe this sentence. “We discovered the evolution, different functions on development, nitrate signaling response, and abiotic stress response between sweet potato and its two diploid relatives.” [Lines 515–517].
4) The article is of great scientific importance. However, the statement in Chapter 5 about the possible increase in sweet potato yields using the data in this article seems controversial.
Response:
We thank the reviewer for pointing this out. The absorption and utilization of nitrogen in sweet potato affects their yield [41]. NLPs are central regulators of nitrate signaling and nitrogen uptake and utilization [16]. Our study analyzed the role of NLPs in nitrate response in sweet potato and provided candidate genes for improving nitrogen use efficiency and increasing yield in cultivated sweet potato.

Reviewer 2 Report
Comments and Suggestions for Authors
The manuscript entitled “Genome-Wide Identification and Expression Analysis of NLPs Family in Sweet Potato and Its Two Diploid Relatives” is an impressive contribution to plant genomics and functional biology. The authors have successfully conducted a comprehensive genome-wide analysis of the NLP gene family in sweet potato and its related diploid species, presenting results that are both scientifically valuable and methodologically sound.
The manuscript is written in a clear, concise, and professional manner, which makes it easy to follow and accessible to a broad scientific readership. The background section provides sufficient context about the importance of NLP gene families in plant growth and development, while the methods are rigorously designed and well described, ensuring reproducibility of the findings.
I find the manuscript suitable for publication in its current format, without the need for revision.
Author Response
Reviewer#2:
The manuscript entitled “Genome-Wide Identification and Expression Analysis of NLPs Family in Sweet Potato and Its Two Diploid Relatives” is an impressive contribution to plant genomics and functional biology. The authors have successfully conducted a comprehensive genome-wide analysis of the NLP gene family in sweet potato and its related diploid species, presenting results that are both scientifically valuable and methodologically sound.
The manuscript is written in a clear, concise, and professional manner, which makes it easy to follow and accessible to a broad scientific readership. The background section provides sufficient context about the importance of NLP gene families in plant growth and development, while the methods are rigorously designed and well described, ensuring reproducibility of the findings.
I find the manuscript suitable for publication in its current format, without the need for revision.
Response:
We thank the reviewer for taking the time to review our manuscript.
